# Shaping Phenolic Resin-Coated ZIF-67 to Millimeter-Scale Co/N Carbon Beads for Efficient Peroxymonosulfate Activation

**DOI:** 10.3390/molecules29174059

**Published:** 2024-08-27

**Authors:** Xin Yan, Yiyuan Yao, Chengming Xiao, Hao Zhang, Jia Xie, Shuai Zhang, Junwen Qi, Zhigao Zhu, Xiuyun Sun, Jiansheng Li

**Affiliations:** Jiangsu Key Laboratory of Chemical Pollution Control and Resources Reuse, School of Environmental and Biological Engineering, Nanjing University of Science and Technology, Nanjing 210094, China

**Keywords:** MOF-derived carbon, porous carbon beads, peroxymonosulfate activation, aminophenol–formaldehyde, tetracycline

## Abstract

Catalytic performance decline is a general issue when shaping fine powder into macroscale catalysts (e.g., beads, fiber, pellets). To address this challenge, a phenolic resin-assisted strategy was proposed to prepare porous Co/N carbon beads (ZACBs) at millimeter scale via the phase inversion method followed by confined pyrolysis. Specially, p-aminophenol–formaldehyde (AF) resin-coated zeolitic imidazolate framework (ZIF-67) nanoparticles were introduced to polyacrylonitrile (PAN) solution before pyrolysis. The thermosetting of the coated AF improved the interface compatibility between the ZIF-67 and PAN matrix, inhibiting the shrinkage of ZIF-67 particles, thus significantly improving the void structure of ZIF-67 and the dispersion of active species. The obtained ZACBs exhibited a 99.9% removal rate of tetracycline (TC) within 120 min, with a rate constant of 0.069 min^−1^ (2.3 times of ZIF-67/PAN carbon beads). The quenching experiments and electron paramagnetic resonance (EPR) tests showed that radicals dominated the reaction. This work provides new insight into the fabrication of high-performance MOF catalysts with outstanding recycling properties, which may promote the use of MOF powder in more practical applications.

## 1. Introduction

The tetracycline antibiotics (TCs) are widely used to treat a number of human and animal diseases, mainly because of their broad-spectrum activity and low cost [1]. Currently, it has been reported that TCs have potential application value in the treatment of epidemic disease (such as flu and COVID-19) [2,3]. However, only 25% of the ingested TC can be metabolized by human body; the rest is released into water bodies. Generally, residual TCs can be absorbed during the growth of flora and fauna, and then taken in by human body through the food chain, leading to health problems such as endocrine disorders, mutagenicity, and antibiotic resistance. Hence, there is an urgent need for a proper method to remove TCs from water bodies [4,5].

Up to now, a number of methods have been used to remove TC from water, such as adsorption [6], biological treatment [7], photocatalysis [8,9], microwave catalysis [10], and advanced oxidation processes (AOPs) [11]. The referenced methodologies exhibit deficiencies such as secondary pollution generation, fiscal infeasibility, diminished industrial efficacy, and diminished efficiency [12]. Among various water treatment technologies, AOPs are recognized for their swift degradation capabilities and potent oxidative performance, which position them as a technology with broad application potential in addressing a variety of environmental pollutants. The sulfate radical (SO_4_^•−^) stands out within SR-AOPs for its longer half-life, which extends from 30 to 40 μs, and its higher redox potential, ranging from 2.5 to 3.1 volts, offering a significant advantage over the hydroxyl radical (·OH). This distinction has propelled SR-AOPs to the forefront of environmental remediation research, where they are explored as an effective strategy for the eradication of micropollutants. The extended half-life and enhanced redox potential of sulfate radicals have not only made SR-AOPs a superior alternative to traditional AOPs but also a leading technique in environmental protection and restoration efforts. The integration of sulfate radicals into AOPs represents an innovative advancement in the field of pollution management, offering a solution that is both effective and environmentally sustainable [13,14].

Developing new catalysts with the superior composition and structure is the key to SR-AOPs. Previous studies have confirmed that cobalt-based materials could efficiently activate peroxymonosulfate (PMS) or persulfate (PS) to generate sulfate radicals due to their remarkable characteristics (e.g., ambient temperature solid oxidant, easy transport and storage, high stability, and high water solubility) [15,16]. However, the inevitable leaching of toxic metal ions from the PS/PMS activation processes remains a challenge for their practical applications. Metal–organic frameworks (MOFs) as superior precursors are widely used for the preparation of carbon-based materials, and many MOF-derived cobalt-based catalysts with diverse structure and composition have been developed [17,18,19]. The organic ligands in MOF plays a role in dispersing and anchoring cobalt during the pyrolysis process, effectively inhibiting the leaching of cobalt in practical applications. Nevertheless, both of MOF powder and its derived carbon materials suffer from easy aggregation and hard transportation, which severely hinder their practical engineering applications. Shaping is one of the approaches to solving the difficulties in the practical application of MOF. Carbon beads, a novel kind of porous carbon material, have received considerable attention. Due to unique physical and chemical characteristics, such as large surface area, low specific density, large controllable inner pore volume, and high mechanical strength, carbon beads have been used in a variety of applications, including drug delivery [20], energy storage [21], and environmental treatment [22]. It was noticed that carbon beads with developed mesoporous structure would facilitate mass transfer efficiency when used as catalysts for eliminating pollutants from environmental media. Moreover, the changeable composition provides the opportunity to apply this novel material for pollutants removal. However, there are currently only relatively limited studies performed on carbon beads as catalysts for pollutant removal. Thus, the development of millimeter-scale carbon beads with desired structural and compositional properties for efficient removal of pollutants is of great significance.

Herein, we developed a facile strategy to fabricate millimeter-sized Co-based carbon beads via phase inversion method followed by pyrolysis. Specifically, ZIF-67, a kind of Co-based MOF, was wrapped by p-aminophenol–formaldehyde (AF) resin, then mixed with polyacrylonitrile (PAN) to prepare polymer beads. After pyrolysis, ZIF-67@AF carbon beads (ZACBs) with distinct porous structure were successfully prepared. The catalytic performance of ZACBs was investigated by PMS activation for degrading organic pollutants. Moreover, the possible decomposition pathways, main reactive oxygen species, and reasonable enhanced mechanism for the ZACB/PMS system were systematically investigated by quenching experiments and electron paramagnetic resonance (EPR).

## 2. Results

The synthetic strategy of the ZACBs is illustrated in Figure 1. First of all, highly uniform ZIF-67@AF nanoparticles were prepared by our previously reported method. Then, the ZIF-67@AF nanoparticles and PAN were dissolved and dispersed to form casting solution. By means of phase inversion using a syringe, the ZAPBs were well prepared. After further carbonization at 900 °C in an inert atmosphere, ZACBs were finally obtained.

### 2.1. Characterizations

Scanning electron microscopy (SEM) was utilized to analyze the morphological characteristics of ZIF-67 and ZIF-67@AF nanoparticles. As depicted in Appendix A, ZIF-67 nanoparticles exhibited a consistent size of approximately 300 nm. In contrast, ZIF-67@AF nanoparticles displayed a uniform size of around 600 nm, which proved the successful introduction of AF layer (Appendix A). After pyrolysis, ZIF-67-C maintained a uniform size with a reduction of about 30%, which is evident in Appendix A. A similar trend was observed for ZIF-67@AF-C, as illustrated in Appendix A. Due to the reduced shrinkage, the formation of a hollow structure was more readily achieved for ZIF-67@AF-C. It was observed that ZIF-67@AF is uniformly distributed throughout the PAN matrix, as illustrated in Figure 1a–c, whereas ZIF-67 tended to aggregate (Appendix A). The likely explanation for this is that ZIF-67@AF was coated with the AF layer, which prevents the particles from clumping together. After pyrolysis at 900 °C, the internal structure of ZACBs (ZIF@AF containing beads) significantly differed from that of ZCBs (ZIF-67 containing beads). As depicted in Figure 1d–f, ZIF@AF retains its original particle shape, whereas the ZIF-67 particles in the original position have disappeared (Appendix A). This may be due to the collapse of ZIF-67 particles by violent contraction during pyrolysis and the interaction of its Co with the -C≡N group of PAN. Throughout the pyrolysis process, ZIF-67 nanoparticles undergo shrinkage from the interior to the exterior, leading to the formation of a well-defined hollow microstructure within the beads. In contrast, the AF layer on the surface of ZIF-67@AF shields the particles from becoming voids. Elemental mapping and line scan results are displayed in Appendix A, confirming the even distribution of C, Co, O, and N elements throughout the ZACBs.

The composition of the samples was subsequently characterized to compare the differences. Firstly, X-ray powder diffraction (XRD) was used to characterize whether the precursors were successfully prepared, from which it can be seen that ZIF@AF as well as its blending with PAN still retained the original lattice of ZIF-67. The results further confirmed that the ZIF-67 and ZIF67@AF nanoparticles were successfully loaded on the PAN polymer substrate (Figure 2a). After pyrolysis, all the ZIF lattices were completely collapsed, and the XRD patterns showed that a strong diffraction peak consisting of the lattice surface with Co (111) existed at ~44° for all the samples. In contrast, as shown in Figure 2b, the diffraction peak width of cobalt shows a broad trend, indicating that the smaller the cobalt nanoparticles in the samples, suggesting that both AF and PAN are effective in preventing the agglomeration of Co nanoparticles during pyrolysis to form more dispersed Co active sites.

Pore structure and specific surface area are also key factors for the performance enhancement of catalytic materials. Based on this, N_2_ adsorption/desorption isotherms were used to determine the specific surface area of the samples and analyze their pore structure. From Figure 3, the results showed that the BET surface areas (S_BET_) and pore volumes (V_pore_) of ZACBs, ZCBs, and ZIF-C were 304.3 m^2^/g and 0.31 cm^3^/g, 293.4 m^2^/g and 0.24 cm^3^/g, 308.6 m^2^/g and 0.29 cm^3^/g, respectively. As can be seen in Appendix A, the presence of PAN greatly reduces the specific surface of ZIF-67, due to the fact that PAN plugs the micropores. However, the similar pore structure obtained after carbonization indicates that the introduction of PAN does not lead to blockage of particle pores.

Mechanical strength is one of the key indexes for engineering applications, and the crushing strength of ZACBs and ZCBs are about 23.59 ± 2.38 N/particle and 9.98 ± 5.83 N/particle, indicating that ZACBs possess better mechanical properties than ZCBs.

### 2.2. Catalytic Performance

The catalytic activities of various reaction systems were evaluated by degrading TC (Figure 4). In the presence of PMS alone, about 43.7% of TC was removed in 120 min, which indicated the weak self-decomposition ability of PMS. Meanwhile, the adsorption of ZACBs catalyst by itself could only remove less than 1% of TC. In contrast, when 10 mg of ZCBs and PCBs were added to the degradation system containing 0.1 g dm^−3^ PMS, about 96.9% and 43.5% of TC were removed in 120 min, respectively, which indicated that the PAN-derived carbon substrate was essentially no contribution. At the same TC concentration and PMS dosage, the catalyst ZACBs removed 99.9% of TC within 100 min, with a constant rate of 0.030 min^−1^ for ZCBs and 0.069 min^−1^ for ZACBs, which was more than 2.3 times that of ZCBs [23].

### 2.3. Effects of pH on TC Degradation

Assessing the impact of various pH levels is vital for gauging the catalyst’s performance in breaking down tetracycline (TC) in real-world applications. The pH at the start of the reaction significantly influences the activation of PMS. Consequently, this research, as illustrated in Figure 5a, examined pH values ranging from 3 to 11 in the degradation system. Notably, at a pH of 3, the system achieved nearly 93% TC degradation in just 120 min. The degradation efficiency of TC increased significantly with the increase of initial pH, and the highest degradation efficiency of the catalyst was achieved at an initial pH of 9. However, when the initial pH was higher than 11, the PMS was activated by alkali and self-decomposed (HSO_5_^-^ react with OH^-^) to generate ^·^OH instead of SO_4_^•-^, which reduced the efficiency of the catalyst [24]. These results indicated that weak alkaline conditions were favorable for promoting the efficiency of the catalyst.

### 2.4. Effects of Temperature on TC Degradation

Beyond the influence of pH, temperature has been scrutinized for its significant role in the degradation mechanism. A selection of temperatures, ranging from 288 K to 308 K, was utilized to assess its impact. As depicted in Figure 5b, an increase in the reaction temperature correlates with a substantial rise in the degradation efficiency of TC. Most notably, at the temperature of 308 K, TC degradation surpasses 80% within a mere 25 min. In line with earlier findings, the kinetic behavior of TC degradation is closely approximated by a first-order kinetic model, represented by the equation that follows (Equation (1)).
(1)$$ln\frac{C_{t}}{C_{0}}=-k_{t}$$
where *k* is the rate constant and *C_t_* and *C*_0_ are the real-time and initial concentration of TC, respectively. In the conducted experiments, the temperatures of the reaction system were meticulously selected as 288 K, 298 K, and 308 K, with corresponding rate constants measured at 0.023 min^−1^, 0.069 min^−1^, and 0.102 min^−1^. The activation energy for the reaction can be deduced from these k values by applying the Arrhenius equation, which is detailed in the equation that comes next (Equation (2)).
(2)$$lnk=-\frac{E_{a}}{RT}+lnk_{0}$$
where *k* and *k*_0_ are reaction rate constant derived from Equation (1), *E_a_* represents the Arrhenius activation energy, *R* is the gas constant, and *T* is the temperature of the system. The Arrhenius activation energy *E_a_* is 55.5 kJ mol^−1^, which is lower than those reported previously [25,26,27], indicating that the ZACBs have a significant catalytic activation in the degradation of TC.

In the process of TC degradation, the catalyst and PMS play a crucial role in the kinetic model. Thus, various catalyst and PMS dosages were investigated in detail. Under the same PMS dosage, the degradation rate of TC rose with the catalyst dose increasing. When the catalyst concentration was adjusted to 50 and 200 mg dm^−3^, the TC degradation efficiency could reach 90% within 120 and 40 min, respectively. These results indicated that increasing the number of active sites within the reaction environment could effectively accelerate the rate of the reaction. Subsequently, various PMS concentrations were explored, as depicted in Figure 5d. Approximately 85% degradation rate was achieved when the PMS concentration was 100 mg dm^−3^. These results showed that ZACBs had a high efficiency for PMS activation. In addition, different concentrations of TC in the system were also examined (Figure 5e). Satisfactorily, approximately 99% of TC could be removed within 80 min when the TC concentration was 20 mg dm^3^, demonstrating that the catalyst had a higher catalytic efficiency at low concentrations of organic pollutants. However, only 88% of TC was completely degraded when the TC concentration came to 40 mg dm^−3^, which might be due to the fact of insufficient PMS dosage, and thus lack of adequate reactive oxygen species for the complete degradation of TC.

### 2.5. Effect of Anions and Organic Matter

In line with prior findings, organic entities such as humic acid (HA) and anions like H_2_PO_4_^-^, HCO_3_^-^ NO_3_^-^, and Cl^-^ are typically present in water bodies [23,24,25]. The radicals formed during the degradation process might engage with these organics and anions, which could lead to a decline in the catalyst’s performance. Hence, it is imperative to evaluate the effects of the aforementioned ions. In the context of natural water systems, Cl^-^, predominantly originating from disinfection byproducts, is more readily oxidized by SO_4_^•-^ to form chlorine radicals (Cl·) with a reduced redox potential. As shown in Figure 5f, a slight suppression phenomenon can be observed when 20 mM H_2_PO_4_^-^, HCO_3_^-^, NO_3_^-^, and Cl^-^ are added into the ZACB/PMS system. Humic acid (HA), known for its high content of carboxyl and phenolic hydroxyl groups, is frequently encountered in natural water systems. The presence of these groups can cause them to interact with the active sites of catalysts that contain graphitic-N, which in turn can lead to a decline in the catalyst’s operational effectiveness [28,29]. When HA was added, the removal efficiency of TC was obviously suppressed and only 75% of TC was removed by the catalyst within 120 min. The results might be due to the generated radicals being consumed by the organic matter.

### 2.6. Degradation Mechanism

Experiments involving radical quenching were carried out to identify the key radical species contributing to TC degradation in the ZACB/PMS system. Methanol (MeOH) is commonly employed to neutralize a range of radicals, including sulfate and hydroxyl radicals, whereas tert-butyl alcohol (TBA) is specifically an effective scavenger for hydroxyl radicals [30,31,32]. As illustrated in Figure 6a, the addition of MeOH at a molar ratio of 1000 to PMS resulted in only approximately 34% TC degradation over 120 min, which points to a radical-mediated pathway in the ZACB/PMS system. In contrast, the introduction of TBA at the same molar ratio led to an 86% reduction in TC, indicating that the presence of ·OH radicals contribute little to the degradation process. The data hint that SO_4_^•-^ might be predominant in TC degradation. According to prior studies, non-radical pathways could involve singlet oxygen (^1^O_2_) and electron transfer, with singlet oxygen being generated from the interaction of the catalyst’s graphitic N active sites with PMS [33,34,35].

As shown in Figure 6a, when 10 mM of NaN_3_ was introduced, it notably reduced the degradation efficiency, with roughly 70% of TC being degraded within 120 min, which points to a contribution from ^1^O_2_ in the degradation of TC. The involvement of ^1^O_2_ was thereby preliminarily validated, and it was deemed significantly accountable for TC breakdown. The graphitic nature of ZACBs, along with the presence of graphitic nitrogen active sites, is instrumental in this reaction. The strong electronegativity of graphite nitrogen, which is attached to carbon atoms in a sp^2^ configuration, draws electrons away from adjacent carbons, creating a positive charge. This scenario is beneficial for the nucleophilic reaction with PMS molecules, which is key to generating ^1^O_2_. In this sequence, the positively charged carbon atoms and electrophilic groups gain electrons from PMS, resulting in the formation of SO_5_^•-^ (as outlined in Equations (3)–(11)) [36]. The outcomes of the radical quenching tests indicate that the graphitic carbon and nitrogen in ZACBs are more likely to promote a reaction that proceeds via non-radical pathways.
Co^2+^ + HSO_5_^-^ → Co^3+^ + SO_4_^•-^ + OH^–^(3)
Co^3+^ + HSO_5_^-^ → Co^2+^ + SO_5_^•-^ + H^+^(4)
HSO_5_^-^ → SO_5_^•-^ + H^+^ + e^−^(5)
SO_5_^•-^ + SO_5_^•-^ → S_2_O_8_^2−^ + ^1^O_2_(6)
SO_5_^•-^ + SO_5_^•-^ → 2SO_4_^•-^ + ^1^O_2_(7)
HSO_5_^-^ + e^−^ → ·OH + SO_4_^2-^(8)
HSO_5_^-^ + e^−^ →SO_4_^•-^ + OH^−^(9)
SO_4_^•-^ + H_2_O → ·OH + SO_4_^2-^ + H^+^(10)
TC + metastable PMS → Intermediates → H_2_O + CO_2_(11)

For a deeper investigation into the primary reactive species in the ZACB/PMS system, the superior technology of electron paramagnetic resonance (EPR) was selected for its proficiency in detecting radicals. The spin-trapping agent DMPO was implemented to trap SO_4_^•-^, and TEMP was used in conjunction for the detection of ^1^O_2_ [37,38,39]. The distinctive peaks presented in Figure 6b suggest that DMPO underwent direct oxidation instead of capturing any radicals. As shown in Appendix A, we can observe a clear 1:1:1 triplet characteristic peak that represents ^1^O_2_, indicating that ^1^O_2_ is involved in the reaction. The hypothesized reaction mechanism for TC degradation within the ZACB/PMS system is outlined in Figure 7.

### 2.7. Recyclability and Stability

Catalytic cycling stability is an important index to evaluate the practical application prospects of catalysts [40,41,42]. As shown in Figure 8, five cycling tests were carried out under the same reaction condition. The catalyst was collected and washed with ethanol and water three times after each cycle, then dried out in an 80 °C oven for 12 h. As observed in Appendix A, the performance of the catalyst ZACBs was found to be on par with others, and under the same macroscopic shaping conditions, the catalytic performance of ZACBs was significantly better than that of other macroscopically shaped catalysts [11,40,43,44,45,46]. ZACBs displayed outstanding degradation performance in each cycle and only a slight decrease could be found at the final recycle, and the removal efficiency of the last recycle was still up to 96.8%, which indicates that ZACBs possess a good reusable characteristic.

## 3. Materials and Methods

### 3.1. Reagents

Cobalt nitrate hexahydrate (Co(NO_3_)_2_·6H_2_O), 2-methylimidazole (2-MeIm), tetracycline (TC), polyvinylpyrrolidone (PVP, K-30), and humic acid (HA) were purchased from Aladdin, (Shanghai, China). Peroxymonosulfate, polyacrylonitrile (PAN, MW = 150,000), and 5,5-dimethyl-1-pyrroline N-oxide (DMPO) were purchased from Sigma-Aldrich, (Shanghai, China). Hydrochloric acid (HCl) was obtained from Sinopharm Chemical Reagent Co., Ltd., (Jiangsu, China), N-dimethylformamide (DMF), sodium hydroxide (NaOH), sodium chloride (NaCl), sodium bicarbonate (NaHCO_3_), sodium dihydrogen phosphate (NaH_2_PO_4_), 4-aminophenol, formaldehyde (37%), ethanol, and tert-butyl alcohol (TBA) were purchased from Nanjing Chemical Reagent Co., Ltd, (Jiangsu, China). Deionized (DI) water was used in all experiments.

### 3.2. Preparation of Porous Carbon Beads

Based on the original reported literature [11,28], some modifications have been made. In a typical experiment, 10.6 g of 2-MeIm and 0.5 mL of HCHO aqueous solution were added to 150 mL of distilled water with stirring for 15 min. Then, 75 mL solution (DI water/ethanol = 2/1) was used to dissolve 0.7 g of Co(NO_3_)_2_·6H_2_O, and 0.3 g of 4-aminophenol was added to the above solution with stirring for 12 h at room temperature. The resultant brown precipitate was collected by centrifugation at 8000 rpm for 5 min and washed with DI water and ethanol three times, respectively. After drying, composite ZIF-67@AF was obtained. ZIF-67 nanoparticles were obtained by the same method without adding formaldehyde and 4-aminophenol, except for collecting by centrifugation at 8000 rpm.

The fabrication of ZIF-67@AF/PAN beads, referred to as ZAPBs, was carried out using a liquid–liquid phase inversion technique, following these steps: Initially, 0.6 g of ZIF-67@AF powder and 0.2 g of PVP were combined and suspended in 6.0 mL of DMF, then subjected to sonication for a duration of 1 h to ensure proper dispersion. Subsequently, 0.8 g of PAN powder was incorporated into the mixture and stirred in a 60 °C water bath for 3 h to achieve a uniform solution. Once air bubbles were eliminated from the solution, it was transferred into a syringe for drop-wise injection at a rate of approximately 40–50 drops per minute. The beads were then allowed to form in water for a period of 24 h, after which they were collected via filtration, rinsed with water, and subsequently dried at a temperature of 60 °C. The ZIF-67/PAN beads, known as ZPBs, were produced by employing an identical procedure. For the creation of PAN polymeric beads, designated as PPBs, the method remained the same but excluded the addition of ZIF-67.

ZACBs, ZCBs, and PCBs were prepared by directly carbonizing ZAPBs, ZPBs, and PPBs in a tubular furnace at a temperature of 900 °C for 120 min under N_2_ atmosphere, respectively. For the comparison, ZIF-67-C was obtained by pyrolysis ZIF-67 powder at 900 °C under N_2_ atmosphere. All the beads were treated by 1 mol dm^-3^ HCl for 24 h.

### 3.3. Characterization

The SEM models FEI 250 (FEI Company, Hillsboro, OR, USA) and JEOL 7800 (Tokyo, Japan) were utilized to scrutinize the structure and surface morphology of the samples. For a deeper look into the internal structure, TEM analysis was carried out on the FEI T20. The crystalline nature of the carbon beads was confirmed by XRD analysis with a BRUKER D8 (Billerica, MA, USA), using a Cu Kα X-ray source with a wavelength of 1.5418 Å, under the conditions of 40 kV and 40 mA. The nitrogen gas adsorption-desorption isotherms were determined at the temperature of liquid nitrogen (77 K) with the aid of a Micromeritics ASAP-2020 analyzer (Norcross, GA, USA). Furthermore, X-ray photoelectron spectroscope (XPS) images were acquired using a PHI Quantera II ESCA System (Chigasaki, Japan), with an Al Kα X-ray source at a photon energy of 1486.8 eV.

### 3.4. Catalytic Experiments

The capability of ZACBs to degrade organic pollutants was evaluated using TC as a model contaminant. The procedure commenced with the addition of 10 mg of ZACBs to 100 cm^3^ of a TC solution at a concentration of 20 mg dm^−3^, followed by stirring at 500 rpm for 30 min to allow for equilibrium between adsorption and desorption. After equilibrium was reached, PMS was added in a quantity of 20 mg to the mixture. An aliquot of 2.0 mL was subsequently extracted for analysis in a cuvette. The spectrophotometric measurement of TC was performed using a UV-Vis spectrophotometer set at 360 nm. The EPR technique, facilitated by a Bruker EMX 10/12 spectrometer (Billerica, MA, USA), was employed to detect radicals, with DMPO serving as the spin-trapping agent to capture ·OH and SO_4_^•-^ radicals.

## 4. Environmental Challenges, Industrialized Capabilities, and Future Considerations

The widespread use of TCs has led to significant environmental challenges, particularly in water bodies. The persistence of TC residues poses a threat to aquatic ecosystems and human health, causing endocrine disorders, mutagenicity, and antibiotic resistance. The environmental challenges are further compounded by the need for effective and eco-friendly remediation strategies that can address the complex nature of pharmaceutical pollution. The development of porous ZACBs demonstrates a promising industrialized capability for the degradation of organic pollutants like TCs. The use of the phase inversion method followed by confined pyrolysis to create these beads showcases a scalable and reproducible approach suitable for industrial applications. The high removal rate of TC within a short time frame and the robust catalytic performance under various conditions highlight the potential for large-scale implementation in wastewater treatment facilities. While the ZACBs showed remarkable performance in laboratory settings, there are several considerations for future research and development. The long-term stability and recyclability of the catalyst under continuous operational conditions need to be evaluated. Moreover, the economic feasibility of scaling up the production of ZACBs and their integration into existing water treatment infrastructure must be assessed. The role of ZACBs in a broader environmental management strategy should also be explored, including the potential for combined treatment methods with other AOPs. Additionally, the environmental impact of the production process itself must be considered, ensuring that the creation of these catalysts aligns with sustainable practices. Lastly, regulatory frameworks and standards for the use of such materials in environmental remediation need to be established or updated to accommodate new technologies. Collaboration between researchers, industry, and policymakers will be crucial to navigate the path from laboratory innovation to practical, large-scale environmental solutions.

## 5. Conclusions

In essence, the fabrication of ZACBs with a pronounced porous structure was successfully conducted at large scale by employing a simple pyrolysis technique on ZIF-67@AF/PAN composite beads. The integration of this unique structure with the relatively dispersed Co active sites led to the ZACBs demonstrating remarkable catalytic activity for the degradation of TC. The insights from the quenching experiments and EPR analysis confirmed that the TC degradation in the ZACB/PMS system was driven by a combination of both radical and non-radical processes, and the former dominated the reaction. The Co leaching experiments indicated that ZACBs exhibit enhanced stability in comparison to ZCBs. Furthermore, the recycling tests suggested that ZACBs had great potential in practical application. In a word, compared to powdered materials, the catalytic performance of ZACBs is in no way inferior, and they are environmentally friendly and easy to apply in practical engineering; compared to shaped materials, their performance far exceeds, and they have excellent reusability. This work expands upon the creation of MOF-derived carbon for the purpose of environmental remediation.

## Data Availability

The authors confirm that all data underlying the finding are fully available without restriction. Data can be obtained after submitting a request to the corresponding/first author.

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
