# Peer review of "Shaping Phenolic Resin-Coated ZIF-67 to Millimeter-Scale Co/N Carbon Beads for Efficient Peroxymonosulfate Activation"

_molecules, 2024, doi:10.3390/molecules29174059_

Round 1

Reviewer 1 Report

Comments and Suggestions for Authors

The research presented is of high quality and the methodology is of a high standard. The characterization of the obtained catalytic materials is quite complete. A comprehensive study has been conducted to assess the catalytic performance, stability, and potential for reuse of the materials in question.

However, there are questions regarding the discussion of the authors' previous works:

1) Why do not the authors cite and discuss their previous work on PMS activation using ZIF-67, dopamine and phenolic resin-based materials - Functional polymers-assisted confined pyrolysis strategy to transform MOF into hierarchical Co/N-doped carbon for peroxymonosulfate advanced oxidation processes, Separation and Purification Technology, 2023?

2) Could you please compare the current result with previous materials based on ZIF-8 (10.1016/j.envres.2021.112618), ZIF-67 (10.1016/j.seppur.2022.122407) and others (10.1016/j.cej.2022.136385).

The authors also investigated the oxidation mechanism of tetracycline and, as far as I understand, showed that the non-radical oxidation pathway predominates. Authors write "In contrast, the introduction of TBA at the same molar ratio led to an 86% reduction in TC, indicating that the presence of ·OH radicals contribute little to the degradation process. The data hint that non-radical mechanisms might be predominant in TC degradation." and "The outcomes of the radical quenching tests indicate that the graphitic carbon and nitrogen in ZACBs are more likely to promote a reaction that proceeds via non-radical pathways".

However in the Conclusion, the authors write “...was driven by a combination of both radical and non-radical processes with the former dominating the reaction”. In fact, in its current form, the phrase means that radical processes dominate. And there is a analogous phrase in the Abstract: "The quenching experiments and electron paramagnetic resonance (EPR) tests shown that radicals dominated the reaction".

The authors need to clarify what processes dominate.

Regarding non-radical pathways, the authors write that probably singlet oxygen is a major contributor to oxidation. Did I understand correctly the phrase “Nonetheless, there was an absence of any signal that would indicate the presence of 1O2”, that despite the fact that you expected to see TEMPO signals after the interaction of TEMP with singlet oxygen, there were none?

On the subject of singlet oxygen, I would like to comment that it is not easy to show the involvement of singlet oxygen in the reaction. Often in the literature, sodium azide is used as an interceptor and TEMP as a precursor of EPR-active TEMPO in singlet oxygen oxidation. However, in reality these are not sufficiently selective mechanistic tests. Azides can block metal centers because they are very good ligands and prevent the oxidant from interacting with the metal, slowing down the reaction. And TEMP in general is oxidized to TEMPO by almost any oxidant, both radicals and various metal peroxocomplexes (in the work Rozantzev E.G. Free Nitroxyl Radicals, 213-214, TEMPO is obtained by oxidizing TEMP with hydrogen peroxide in the presence of Na2WO4, which definitely does not generate singlet oxygen). Therefore, TEMP is non-selective which was also mentioned in the paper Moan, J., Wold, E. (1979). Detection of singlet oxygen production by ESR. Nature, 279(5712), 450-451.

More reliable ways to establish the involvement of singlet oxygen are to compare the course of the reaction in deuterated and non-deuterated solvent, and specific substrates that generate endoperoxides upon reaction with singlet oxygen, such as dimethylanthracene (there are also water-soluble substrates).

In conclusion, the article is deemed to be of merit and is recommended for publication.

Comments on the Quality of English Language

There are a few awkward phrases and typos, I recommend carefully double-checking the manuscript. Examples:

1) "Among virous technologies, AOPs" 40

2) "a slightly suppression phenomenon can be found" 227

3) "non-radical processes. with the former dominated the reaction" 352

Reviewer 2 Report

Comments and Suggestions for Authors

In this MS, Yan et al developed a facile strategy to fabricate millimetre-sized Co-based carbon 78 beads via the phase inversion method followed by pyrolysis and studied its catalytic performance.

The MS is indeed well-prepared and the reported results are noteworthy.

The reviewer thinks it can be published after addressing some comments.

1. The introduction should be enriched in some cases: a. regarding the TC removal, some common methods and their pros and cons should be added. Preparation and characterizations of TiO2/ZnO nanohybrid and its application in photocatalytic degradation of tetracycline in wastewater. b. More discussion about the advantages of using AOP techniques.Investigation of H2O2/UV advanced oxidation process on the removal rate of coliforms from the industrial effluent: A pilot-scale study. c. Add some recent works in a tabulated format summarizing their outcomes.

2. Highlight the novelty of this work in comparison with others.

3. The Materials and methods section should move to 2 and the results should be followed.

4. The preparation procedure should be cited.

5. All equations should be cited using relevant works. Nitrate removal performance of different granular adsorbents using a novel fe-exchanged nanoporous clinoptilolite. Also, can the authors study more kinetic modes?

6. The environmental challenges, industrialized capabilities, and future considerations should be discussed in a separate section before the conclusion. 

Reviewer 3 Report

Comments and Suggestions for Authors

The manuscript may be published with a major revision. The following points should be considered before publication:

-          Line 52 - sustainable. – delete point

-          Line 56 - In the Introduction part to improve the quality of the manuscript include the following article (doi 10.2298/JSC190313035M) and explain advantages of persulfate in comparison with peroxymonosulfate (it is a solid oxidant at ambient temperature, its transport and storage are easy, it has high stability, and high water solubility)

-          Line 92 - oC - use proper symbol with ° in superscript - °C

-          Line 168 - persulfate (PMS) – it should be peroxymonosulfate, abbreviation for persulfate is PS

-          Use SI units according to journal instructions for authors – make changes in the entire manuscript – for example cm3 instead of mL, mol dm–3 instead of M, mg dm–3 instead of mg L-1, etc.

-          Line 192 end equation 1 - Ct and C0 should be small italic. Make changes to the entire manuscript.

-          Line 195 – k and all parameters should be italic. Make changes to the entire manuscript.

-          Line 206 - since the efficiency degradation rate of TC increases with the increase in the concentration of the catalyst, in the examined concentration range from 50 to 200 mg dm–3, examine what happens with a further increase in the catalyst concentration, i.e. does saturation occur. The same situation is for PMS concentration influence experiments. Investigate degradation efficiency of TC for PMS concentrations higher than 400 mg dm–3.

-          In order to strengthen the quality of the manuscript include the following article (doi 10.2175/106143017X15131012152924) and all applied kinetic models. Kinetics of the degradation process must be discussed.

-          Add a comparation of degradation efficiency of tetracycline by the results from the literature, such as 10.1016/j.seppur.2024.128673, 10.1016/j.seppur.2024.128646, 10.1016/j.seppur.2024.128648, etc. For comparison choose the materials from the literature with the highest efficiency.

-          Line 294 - Sino-pharm - it should be Sinopharm

-          Line 317 - degrees Celsius – use °C as in the rest of the manuscript

-          Line 336 – milligrams – use mg

-          Line 337 – milliliters – use cm3

-          Supplementary material – Fig.S5 (b) instead of (d)

Round 2

Reviewer 2 Report

Comments and Suggestions for Authors

Equations 1 and 2 are uncited. The MS can be accepted after addressing this comment.

Reviewer 3 Report

Comments and Suggestions for Authors

The authors answered all my questions. The manuscript can be accepted in present form.